# Histone Deacetylases as Epigenetic Targets for Treating Parkinson’s Disease

**DOI:** 10.3390/brainsci12050672

**Published:** 2022-05-21

**Authors:** Yan Li, Zhicheng Gu, Shuxian Lin, Lei Chen, Valentina Dzreyan, Moez Eid, Svetlana Demyanenko, Bin He

**Affiliations:** 1State Key Laboratory of Functions and Applications of Medicinal Plants, Engineering Research Center for the Development and Application of Ethnic Medicine and TCM (Ministry of Education), Guizhou Provincial Key Laboratory of Pharmaceutics, School of Pharmacy, School of Basic Medical Science, Guizhou Medical University, Guiyang 550004, China; yanli@gmc.edu.cn (Y.L.); guzhicheng0520@163.com (Z.G.); linshuxian365@163.com (S.L.); leichen@gmc.edu.cn (L.C.); 2Laboratory of Molecular Neurobiology, Academy of Biology and Biotechnology, Southern Federal University, Stachki Ave. 194/1, 344090 Rostov-on-Don, Russia; dzreyan@sfedu.ru (V.D.); moez1995.mae@gmail.com (M.E.)

**Keywords:** Parkinson’s disease, epigenetic targets, histone deacetylases, inhibitors, PROTACs

## Abstract

Parkinson’s disease (PD) is a chronic progressive neurodegenerative disease that is increasingly becoming a global threat to the health and life of the elderly worldwide. Although there are some drugs clinically available for treating PD, these treatments can only alleviate the symptoms of PD patients but cannot completely cure the disease. Therefore, exploring other potential mechanisms to develop more effective treatments that can modify the course of PD is still highly desirable. Over the last two decades, histone deacetylases, as an important group of epigenetic targets, have attracted much attention in drug discovery. This review focused on the current knowledge about histone deacetylases involved in PD pathophysiology and their inhibitors used in PD studies. Further perspectives related to small molecules that can inhibit or degrade histone deacetylases to treat PD were also discussed.

## 1. Introduction

Parkinson’s disease, the second most common neurodegenerative disease, is characterized by motor (e.g., bradykinesia, rigidity, and tremor) and non-motor (e.g., constipation, hyposmia, depression, cognitive decline, and sleep alterations) signs and symptoms, which affects roughly 1% of elders over the age of 60 years [1]. With the aging of the population, the incidence of PD rises. In some countries, especially in North America and Western Europe, the risk of Parkinson’s disease is annually increasing [2]. In China, according to the newly-released seventh national census, the population aged 60 and over has reached 264.02 million, which may also result in the increasing risk of neurodegenerative diseases such as Parkinson’s disease [3]. Additionally, the incidence of PD is estimated to be doubled worldwide by 2040 [4,5,6]. In front of this global threat, the exact pathogenesis of PD remains unclear, but it may link to both genetic and environmental factors. Since two classical hallmarks of the loss of dopaminergic neurons from the substantia nigra pars compacta (SNpc) and the accumulation in Lewy bodies (LB) from the aggregates of α-synuclein have been established, the agents for “symptomatic” (alleviating the features of the condition) treatment of PD have been used in clinic or investigated in clinical trials. The drugs approved by the U.S. Food and Drug Administration (FDA) for treating Parkinson’s disease can be roughly classified into dopaminergic system drugs, serotonergic system drugs, cholinergic system drugs, and others [7]. Dopaminergic system drugs include aromatic amino acid decarboxylase (AADC) inhibitors, catechol-O-methyltransferase (COMT) inhibitors, monoamine oxidase B (MAO-B) inhibitors, dopamine transporter (DAT) inhibitors, and dopamine receptor (DR) agonists; serotonergic system drugs are 5-hydroxytryptamine (5-HT) 2A and 2C receptor antagonists; cholinergic system drugs are mainly muscarinic acetylcholine receptor (mAChR) antagonists; adenosine receptor (A2A) inhibitors and glutamate receptor (NMDA) antagonists are considered as other drugs for treating PD [8,9,10,11]. These treatments only provide short-term relief from PD symptoms, but no treatment has been successfully converted into a clinically disease-modifying treatment yet [1,12,13,14]. Therefore, there is still an urgent need to target other potential mechanisms to develop effective treatments that can modify the course of Parkinson’s disease to improve PD patients’ quality of life.

The term “epigenetics” is used to describe the regulatory mechanism involving the modification of the expression levels of certain genes without altering their DNA sequence, generally including DNA methylation, histone acetylation, and RNA modifications [15]. Since epigenetic changes occur in most diseases, epigenetic targets in drug discovery have received much attention in recent years. In particular, histone acetylation can cause a relaxation of chromatin to allow transcriptional factor access to the DNA and thus promote gene transcription. Among epigenetic targets, histone deacetylases have been intensively studied (for searching the clinical trials targeting histone deacetylase, please go to the website: https://www.clinicaltrials.gov/, accessed on 20 May 2022) because of their capability to remove acetyl groups from the lysine residues of histone and nonhistone substrates and thus regulating chromatin remodeling or the expression of certain genes [16]. As shown in Table 1, HDACs have been divided into four classes, in which Class I (HDAC1, 2, 3, and 8), Class IIa (HDAC4, 5, 7, and 9), Class IIb (HDAC6 and 10), and Class IV (HDAC11) are zinc-dependent deacetylases, while Class III (SIRT 1-7) are nicotinamide adenine dinucleotide (NAD^+^)-dependent deacetylases (also called sirtuins) [17,18]. Epigenetic modulation by HDAC inhibition has proven to be beneficial for treating cancer [19]. To date, the U.S. Food and Drug Administration (FDA) has approved four HDAC inhibitors to be used as a monotherapy to treat several cancers, including Vorinostat (SAHA) [20], Romidepsin (FK228) [21], and Belinostat (PXD-101) [22] for treating cutaneous T-cell lymphoma (CTCL) or peripheral T cell lymphoma (PTCL) and Panbinostat (LBH-589) [23] for the treatment of multiple myeloma. Additionally, Chidamide was approved by the National Medical Products Administration (NMPA) in China for the treatment of PTCL [24]. Despite these successful applications of HDAC inhibitors in cancer treatment, recent studies have shown that the inhibition of histone deacetylases could be extended as a potential way of treating neurodegenerative diseases such as Parkinson’s disease [25,26,27,28,29].

In this review, we focused on the latest knowledge regarding the possible role of histone deacetylase in the pathophysiology of Parkinson’s disease, and we also examined the present state of pan- or selective inhibitors against histone deacetylase with the therapeutic potential for treating Parkinson’s disease. Additionally, we discussed the future perspectives related to histone deacetylases as epigenetic targets for treating Parkinson’s disease.

## 2. Histone Deacetylases Involved in Parkinson’s Disease Pathophysiology

Although we cannot fully understand the pathogenesis of PD, it is quite clear that the loss of dopaminergic neurons [30,31,32], α-synuclein aggregation [33], metal ion accumulation [34], and oxidative stress [35] play crucial roles in PD pathogenesis (Figure 1) [36]. On the other hand, transcriptional dysregulation often occurs in the progression and development of many neurodegenerative central nervous system (CNS) diseases, including Parkinson’s disease [37]. On these grounds, histone deacetylases, as an important group of epigenetic targets associated with transcription regulators, play a significant role in chromatin remodeling and gene expression by regulating the status of histone acetylation, probably involved in PD pathogenesis (Figure 1). Among 18 mammalian histone deacetylases, there are different roles for each histone deacetylase isoform, displaying either a neuroprotective or neurotoxic effect [38]. For example, HDAC1 has been proven to be a critical factor for protecting neurons from DNA damage in several neurodegenerative diseases. Furthermore, the activation of HDAC1 promoted neuroprotective activity in human neuronal models of neurodegeneration [39]. For Class III HDAC (Sirtuins), SIRT1 and SIRT5 are also known to be neuroprotective, while SIRT2, SIRT3, and SIRT6 are known to be neurotoxic [40,41,42]. A study of the localization and expression of SIRT1, 2, 6 and plasticity-associated proteins in the recovery period after ischemic stroke in mice showed that there was an increase in the levels of SIRT1 and SIRT2 during this recovery period. The increase in SIRT1 was associated with an increase in synaptic plasticity proteins, while the increase in SIRT2 was associated with α-tubulin acetylation, which may reduce neurite motility. Moreover, it was shown that SIRT1, SIRT2, and SIRT6 were not involved in ischemic stroke-induced penumbra cell apoptosis [43]. Most HDACs are expressed in neurons. For example, HDAC1 is expressed primarily in neurons, and its interaction with HDAC3 in neurons may promote toxicity, while HDAC4 in neuronal nuclei appears to be associated with neuronal cell death [38]. On the other hand, there is some evidence that most Class II HDACs demonstrated neurotoxic effects [44]. Our inhibitory analysis using HPOB [45] and tubastatin A [46], selective HDAC6 inhibitors, confirmed the involvement of this histone deacetylase in neurodegeneration. In terms of the expression levels of HDACs in PD brain samples, there are conflicting results from two separate studies. One study showed a decrease in protein levels of HDAC1, HDAC2, HDAC6, and SIRT1 in PD brain samples compared to those in controls [47]. Another study, on the contrary, demonstrated that there were no significant differences in the expression of Class I HDACs, Class II HDACs, and Class III (SIRT1 and SIRT2) in PD SNpc, as compared with age-matched controls [48]. However, a transcriptome study of clinical PD samples gave a 1.6-fold higher level of HDAC6 expression and a 1.65-fold higher level of histone acetyltransferase1 (HAT1) expression [49,50], which is consistent with the increasing acetylation levels of H3 histone reported in the above studies [47,48]. Overall, HDAC1 is expressed throughout the brain and most abundantly expressed in the cerebellum, while HDAC2 is expressed higher in the brain than HDAC1. However, HDAC3 is the most highly expressed in the brain among Class I HDACs, and the expression of HDAC8 in the brain, the only one of Class I HDACs, is still not clear. For the rest of HDACs, high expressed HDAC4-6 are found in the brain, while HDAC7-10 are relatively expressed at low levels in the brain, and HDAC11 remains to be disclosed [38]. According to the current acknowledge, we may safely say that HDAC1-3 and HDAC6 are neurotoxic while HDAC7, HDAC9, and HDAC10 are neuroprotective regarding their expression in the brain.

Recent studies revealed that HDAC5 and HDAC9 are co-expressed with nigrostriatal dopaminergic markers, not only in the human SN but also in dopaminergic neurons in adult mouse substantia nigra (SN). Silencing the expression of HDAC5 or HDAC9 by siRNAs can promote neurite growth in SH-SY5Y (a human neuroblastoma cell line) cells. Pharmacological inhibition by their inhibitors, MC1568 or LMK235, can protect rat dopaminergic neurons in the presence of the neurotoxin, 1-methyl-4-phenylpyridinium ion (MPP^+^) or α-synuclein [51,52]. Further studies showed that intraperitoneal (IP) administration of MC1568 in rats reduced HDAC5 levels in the nucleus accumbens and thimerosal-induced apoptosis in the rat prefrontal cortex. More importantly, MC1568 via IP treatment can exert neuroprotective effects in a 6-hydroxydopamine-induced (6-OHDA-induced) rat model of PD [53,54]. Furthermore, there is growing evidence that HDAC4 exerted neurotoxicity by its nuclear accumulation and involvement in both Lewy and Marinesco bodies in models of Parkinson’s disease [55,56,57]. Class IIa HDACs exerted effects on neuronal survival and on axonal growth via nucleocytoplasmic shuttling [58,59], and pharmacological inhibition with Class IIa-specific inhibitors protected dopaminergic neurons from neurotoxin- and α-synuclein-induced degeneration in cellular models of Parkinson’s disease [52,60]. Current results indicated that Class IIa HDACs had a detrimental effect on dopaminergic neurons, and the inhibition or cytoplasmic shuttling of Class IIa HDACs (at least HDAC4 or HDAC5) might improve the neuroprotective effects in PD [51,60].

Although dopaminergic neurons in brains have been shown to express both Class IIb HDACs (HDAC6 and HDAC10) [61], most studies have been focused on HDAC6, and little is known about HDAC10 [44]. HDAC6 is highly expressed in Lewy bodies in PD patients’ brain sections, indicating that HDAC6 may play a key role in the clearance of those misfolded and aggregated proteins [62,63]. In an α-synuclein-induced *Drosophila* model of PD, HDAC6 was shown to interact with oligomeric α-synuclein directly and thus to be involved in Lewy bodies formation [64]. In a lactacystin-induced mouse model of PD, the HDAC6 expression level was selectively increased in the SN region of the brain, resulting in perinuclear inclusion bodies that were structurally similar to aggresomes. However, treatment with trichostatin A (TSA), a pan-HDAC inhibitor, increased α-synuclein oligomer levels in the SN and associated behavioral deficits, while treatment with tubacin, a selective HDAC6 inhibitor, exacerbated lactacystin-induced cell death in primary neuron cells, suggesting that the deacetylase activity of HDAC6 is essential for its protective effects on dopaminergic neurons in this model [65]. Since R1441C and Y1669C mutations in leucine-rich repeat kinase 2 (LRRK2) are the pathogenic LRRK2 mutants in sporadic PD, a *Drosophila* model that expressed the LRRK2 mutants (R1441C and Y1669C) has been used to investigate the role of HDAC6 in PD [66,67,68,69,70,71]. In this *Drosophila* model, HDAC6 knockdown can effectively restore axonal transport and motor behavior. Moreover, treatment with TSA was also shown to restore axonal transport and improve the associated behavioral deficits [71]. Similar results were obtained by treatment with tubastatin A (a specific HDAC6 inhibitor) in an MPP^+^-induced Zebrafish model of PD [72]. However, there were contrast results that treatment with tubastatin A did not rescue impairments in spontaneous movements or sensorimotor reflexes in an MPP^+^-induced Zebrafish model of PD, while treatment with 4-phenylbutyrate, a pan-HDAC inhibitor, improved those PD symptoms [72]. Another animal model was a 6-OHDA-induced mouse model of PD. In this PD model, HDAC6 deacetylation of peroxiredoxin1 (Prx1) and peroxiredoxin2 (Prx2) contributes to oxidative injury in PD [73]. The pharmacological inhibition of HDAC6 with tubastatin A increased the acetylation of Prx1 and Prx2, reduced reactive oxygen species (ROS) production, and alleviated dopaminergic neurotoxicity [73]. Overall, in several animal models of PD, HDAC6 seems to have a crucial role in PD pathogenesis and progression.

Among Class III HDACs (sirtuins), different subtypes may also have different roles in neuronal survival and neurodegeneration [40]. In general, SIRT1 and SIRT5 may have neuroprotective effects, while SIRT2, SIRT3, and SIRT6 may have neurotoxic effects [74,75,76]. Especially, SIRT1 can bind with several transcription factors, e.g., NF-κB, p65, retinoic acid receptor β (RARβ), forkhead box O (FOXO), and peroxisome proliferator-activated receptor-gamma coactivator 1α (PGC1α) associated with PD, and the activation of SIRT1 can maintain mitochondrial quantity and function and further reduce α-synuclein aggregation [77,78,79]. On the contrary, although SIRT2 can directly bind with and deacetylate FOXO3a upon caloric restriction and oxidative stress [80], significant overexpression of SIRT2 was found in Parkinson’s patients compared to normal group [81]. Furthermore, the substrates of SIRT2, such as α-tubulin and p53, may be involved in neurotoxicity and neurodegeneration [82,83,84]. Most importantly, SIRT2 was found to regulate α-synuclein-mediated toxicity in vitro and in vivo models of PD, although the exact mechanism remains uncertain [80,85,86].

## 3. Histone Deacetylase Inhibitors for Parkinson’s Disease

Since an imbalance of lysine acetylation in histones and nonhistone proteins, along with several histone deacetylase isoforms, may be involved in the pathogenesis of PD, scientists have put much effort into developing histone deacetylase inhibitors in PD studies (Figure 2) [16,87,88,89,90]. Valproic acid, as a pan-HDAC inhibitor, was initially shown to protect dopaminergic neurons and attenuate lipopolysaccharide (LPS)-induced dopaminergic neurotoxicity in cells [91,92]. The following studies have shown that valproic acid has the protective effects on the nigrostriatal dopamine system in several PD models, including 1-Methyl-4-phenyl-1, 2, 3, 6-tetrahydropyridine-induced (MPTP-induced) mice, lactacystin-induced mice, leucine-rich repeat kinase (LRRK2) R1441G transgenic mice and 6-OHDA-induced mice [93,94,95,96]. Sodium butyrate, another pan-HDAC inhibitor and structurally similar to valproic acid, was found to have beneficial effects in 6-OHDA-induced neurotoxicity and behavioral abnormalities [97,98]. Against salsolinol-induced cytotoxicity in SH-SY5Y cells, sodium butyrate demonstrated neuroprotective effects [99]. In a rotenone-induced *Drosophila* model of PD, sodium butyrate could improve locomotor impairment and early mortality [100]. Additionally, a simple derivative of butyrate, phenylbutyrate, also exerted neuroprotective effects in an MPTP-induced mice model of PD [101,102]. TSA, one of the most potent pan-HDAC inhibitors, attenuated manganese chloride-induced cell death and apoptosis in PC12 cells [103]. However, a single treatment with TSA resulted in enhanced cell death and increased apoptosis in MPTP- or rotenone-induced cell models of PD [104]. In MPP^+^- or MPTP-induced cells and mice models of PD, TSA could protect the integrity of mitochondria and neuron cell survival [105,106]. Suberoylanilide hydroxamic acid (SAHA), one of the approved HDAC inhibitors for treating cancers, has been studied for its potential new application in the therapy of PD. SAHA was found to protect dopaminergic neurons from neurotoxin-induced damage in in vitro Parkinson’s disease models [107]. These results suggested that pan-HDAC inhibitors may have a potential neuroprotective role in PD [108]. On the other hand, selective inhibitors have also been investigated in the treatment of PD. RGFP109, a selective inhibitor of HDAC 1 and 3, alleviated *L*-3,4-dihydroxyphenylalanine-induced (*L*-DOPA-induced) dyskinesia in the MPTP-lesioned marmoset [109]. An HDAC1/2 dual inhibitor, K560, and its derivative, K-856, were shown to have protective effects in MPP^+^- or MPTP-induced in vitro and in vivo models of PD [110,111]. By the comparison of MS-275 (a Class I HDAC inhibitor) and MC-1568 (a Class II HDAC inhibitor), it was found that inhibition of Class I HDAC attenuated polychlorinated biphenyls-induced (PCBs-induced) neuronal cell death by preventing HDAC3 binding [112]. However, LMK235, a Class IIa HDAC inhibitor (especially for HDAC4/5), exerted neuroprotective effects in MPP^+^-treated SH-SY5Y cells or cultured DA neurons. Furthermore, similar results for LMK235 were obtained relating to the protection of axonal degeneration induced by the overexpression of wild-type or A53T-mutant α-synuclein in both SH-SY5Y cells and cultured DA neurons [52].

Among 18 different members of the histone deacetylase family in humans, sharing the common deacetylation substrate, microtubule-associated tubulin, HDAC6 and SIRT2 are predominant in the cytoplasm and play significant roles in a variety of neurodegenerative disorders, which make these two isoforms become most attractive targets for treating neurodegenerative diseases, including Parkinson’s disease. In particular, HDAC6 is required for the centrosome recruitment and dispersion of parkin, which is linked to PD [113]. Tubastatin A, as an HDAC6 inhibitor, was shown to protect dopaminergic neurons in a rat model of PD through the reduction in the α-synuclein level and neuroinflammation, the promotion of chaperone-mediated autophagy, and the elimination of other PD-related pathological pathways [114,115]. Notably, HGC is a newly developed HDAC6 inhibitor, which improves dopaminergic neuron viability and attenuates behavioral defects in PD-modeling cells and animals by the accumulation of K28 acetylation in NADH-ubiquinone-oxidoreductase flavoprotein 1 (NDUFV1), and thus maintaining mitochondrial integrity and functions [116]. Interestingly, venlafaxine, in the clinical treatment of depression, was found to inhibit HDAC6 expression and then enhance α-synuclein clearance via the activation of the ubiquitin–proteasome system (UPS) and autophagy in a rotenone-induced mice model of PD [117]. Additionally, several probes derived from HDAC6 selective inhibitors may provide powerful tools for investigating HDAC6 in various physiological and pathological conditions. By the combination of Tubastatin A and 1, 8-naphthalimide-based fluorophore, a fluorescent HDAC6 selective inhibitor **6b** has been developed, demonstrating the capability of labeling and visualizing HDAC6 in inclusion bodies and aggresomes [118]. From a brain penetrant HDAC6 inhibitor (Bavarostat), [^18^F] Bavarostat, an ^18^F labeled analog, was achieved and demonstrated a utility for the mapping of HDAC6 in the living brain [119]. Another ^18^F-labeled tetrahydroquinoline derivative, [^18^F] **2**, based on the HDAC6-selective inhibitor, SW-100, was obtained and was also shown to be of potential use for brain HDAC6 imaging by positron emission tomography (PET) [120]. AGK2 was the first SIRT2 inhibitor reported in the PD study, which rescued α-synuclein toxicity and protected against dopaminergic cell death both in vitro and in a *Drosophila* model of PD [48,86]. AGK2 was found to decrease H_2_O_2_-induced apoptosis of differentiated PC12 cells, suggesting that SIRT2 plays a significant role in oxidative stress-induced cell death [121]. Interestingly, SIRT2 inhibition by AGK2 also exerted the neuroprotective effects in ischemic stroke by the downregulation of AKT/FOXO3a and MAPK pathways [122]. AK-7, as another SIRT2 selective inhibitor, was observed to ameliorate α-synuclein toxicity in vitro and significantly improve behavior abnormality and neurochemical deficits in an MPTP-induced mice model of PD [123,124,125,126,127]. ICL-SIRT078, as a substrate-competitive SIRT2 inhibitor, demonstrated a significant neuroprotective effect in a lactacystin-induced mice model of PD [128]. By a similar strategy, recently developed peptide YKK (ε-thioAc) AM can structurally mimick SIRT2 substrates, and thus competitively inhibit the activity of SIRT2. YKK (ε-thioAc) AM further showed neuroprotective effects in the PC12 cells model of PD [85]. The scaffolds of 3-(N-arylsulfamoyl) benzamide and 5-((3-amidobenzyl) oxy) nicotinamides may provide other options to explore more SIRT2 inhibitors as a potential therapy for Parkinson’s disease [129,130].

## 4. Summary and Further Perspectives

Parkinson’s disease (PD) is increasingly becoming a global threat and burden. The risk of Parkinson’s disease has increased worldwide, especially for elderly people above the age of 60. Current treatments can only alleviate the PD symptoms, but none of the treatments are disease-modifying [1,12,13,14]. Does targeting other potential mechanisms in order to develop effective treatments create an opportunity for modifying Parkinson’s disease? To our knowledge, this is still a key question awaiting scientists to address and requires further research. Epigenetic therapy is a relatively new concept, and epigenetic targets have attracted much attention in drug discovery for dealing with many human diseases [15]. In this context, our attention was focused on an important group of epigenetic targets, histone deacetylases, including HDACs and sirtuins, for the treatment of Parkinson’s disease (Figure 3). There are 18 histone deacetylases in humans that may play different roles in PD. From the current knowledge, Class I HDACs seem to be neuroprotective, while Class II HDACs appear to be neurotoxic [38,39,40,41,42]. In particular, the activation of HDAC1, one of Class I HDACs, was beneficial for treating neurodegenerative diseases, including PD. Similarly, the activation of SIRT1, one of Class III HDACs, was helpful in reducing α-synuclein aggregation in PD. Therefore, the activation of HDAC1 and SIRT1 could be an interesting strategy for treating PD [39,72,79,88,103,110,111]. On the contrary, Class IIa HDACs, such as HDAC4 and HDAC5, were shown to have detrimental effects on dopaminergic neurons in cellular models of PD. HDAC6, one of Class IIb HDACs, is the most intensively studied member of the histone deacetylase family in PD. HDAC6 was found to interact with α-synuclein directly and is involved in Lewy bodies formation, implying that HDAC6 may relate to PD pathogenesis and progression [114]. Moreover, HDAC4 may also have a similar role in LB formation [52]. SIRT2, one of Class III HDACs, was found to be significantly overexpressed in PD [40,41,42,48], and its regulatory substrates are also involved in neurotoxicity and neurodegeneration [82,83,84].

Although these studies have disclosed that some isotypes of histone deacetylases may be suitable epigenetic targets for treating PD, the amount of information about the roles of histone deacetylase in PD is relatively limited, and more extensive research is required to validate their roles and regulatory mechanism in PD by using their inhibitors. Actually, both HDAC inhibitors and sirtuin inhibitors were reported to have neuroprotective effects on midbrain dopamine neurons (Figure 3). Moreover, they have been studied in cell and animal models of PD. Initially, many studies have reported the neuroprotective effects of some pan-HDAC inhibitors, such as valproic acid, sodium butyrate, phenylbutyrate, TSA, SAHA, and others, in different animal models of PD [91,92,93,94,95,96,97,98,99,100,101,102,103,104,105,106,107,108]. Since glutamate, as a neurotransmitter, is highly associated with PD, there are reports on the neuroprotective mechanism of TSA preventing glutamate-induced toxicity in the medium of primary cultured astrocytes treated with MPP^+^ [131]. However, most of these pan-HDAC inhibitors have less inhibitory potency and poor selectivity, which raises concerns about the efficacy and toxicity of using them for PD treatment. There is evidence for the neuroprotective potential of isotype-selective inhibitors of human histone deacetylases in in vivo models of PD, although the neuroprotective effects of class II inhibitors are still under question. At least, SIRT2 inhibitor showed neuroprotective effects not only in an α-synuclein-induced *Drosophila* model of PD but also in an MPTP-induced mice model of PD [48,85,86,121,122,123,124,125,126,127,128,129,130]. Additionally, HDAC6 inhibitors also showed neuroprotective potentials in several mice models of PD [113,114,115,116,117,118,119,120].

Some isotypes of human histone deacetylases may have neuroprotective roles in PD, and nicotinamide, a pan-Class III HDAC inhibitor against all SIRT1-7, exacerbated the neurotoxic effects in a lactacystin-induced rats model of PD [132]. There is an urgent need for more comprehensive research to explore the exact role of each isotype of histone deacetylases in PD by appropriate isotype-specific inhibitors. It will be crucial to determine the role and regulatory mechanism of each isotype of histone deacetylases in PD to have isotype-selective inhibitors with higher therapeutic efficacy and lower toxicity in the clinical treatment of PD in the future. Future studies on the development of isotype-selective inhibitors, especially against HDAC6 and SIRT2, with more inhibitory potency, high selectivity, and the ability to penetrate the blood-brain barrier, will facilitate further exploration of their therapeutic potentials in PD. For example, the brain-penetrant histone deacetylase inhibitor, RGFP109, attenuates *L*-DOPA-induced dyskinesia in the MPTP-lesioned marmoset [133].

Currently, the emerging proteolysis targeting chimera (PROTAC) technology has been widely used to degrade selectively aberrant target proteins, including epigenetic proteins, which have potential clinical applications for treating many human diseases, including neurodegenerative diseases [134,135]. As compared to traditional inhibitors, PROTACs may have several advantages for the treatment of PD. For example, PROTACs have higher specificity and can avoid the potentially toxic effects induced by traditional inhibitors. Moreover, PROTACs can exert chemically-induced degradation of a target protein and thus interfere in the enzymatic and non-enzymatic function of a target protein. Moreover, PROTAC molecules are released once the degradation cycle is completed and then can be further reused for another degradation cycle until the target protein is eliminated. Therefore, PROTACs only need to be administered in very low concentrations, which is sufficient to degrade a target protein. These advantages make PROTACs a highly effective way for the degradation of those proteins associated with the pathophysiology of various neurodegenerative disorders, including PD. Very recently, some PROTACs with high selectivity and lower cytotoxicity have been developed for the degradation of HDAC6 or SIRT2 [132,136,137,138,139,140,141,142,143,144]. However, none of them have been tested for their therapeutic potential in treating PD. Future investigations of these PROTACs targeting epigenetic proteins to treat PD would be exciting to explore. Although most clinical trials of HDAC inhibitors are related to the treatment of cancers, some clinical studies about HDAC inhibitors with carboxylic acids as the ZBG groups for the treatment of neurodegenerative diseases are ongoing [25]. As we can easily imagine, more selective inhibitors, multitarget HDAC-based inhibitors, or PROTACs, may facilitate clinical trials for neurodegenerative diseases, including PD.

## Figures and Tables

**Figure 1 brainsci-12-00672-f001:**
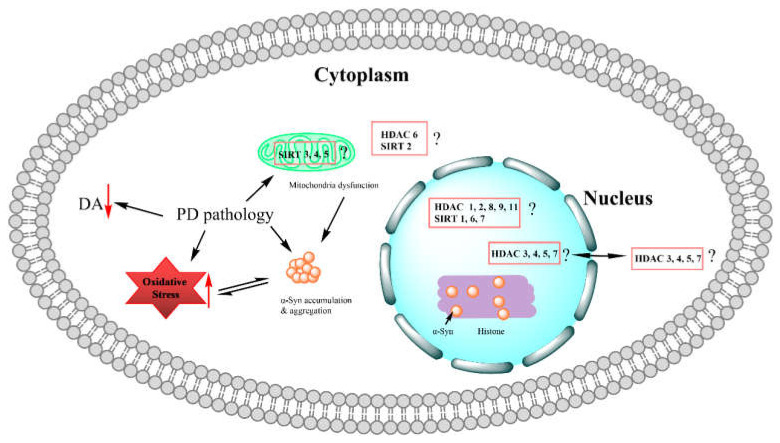
PD pathology and localizations of histone deacetylases. Current knowledge about the PD pathology mainly includes dopamine (DA) decrease, α-synuclein aggregation, oxidative stress, and mitochondrial dysfunction; whether histone deacetylases are linked with this pathogenesis in PD still remains to be explored and illustrated. This leaves a question mark for every histone deacetylase.

**Figure 2 brainsci-12-00672-f002:**
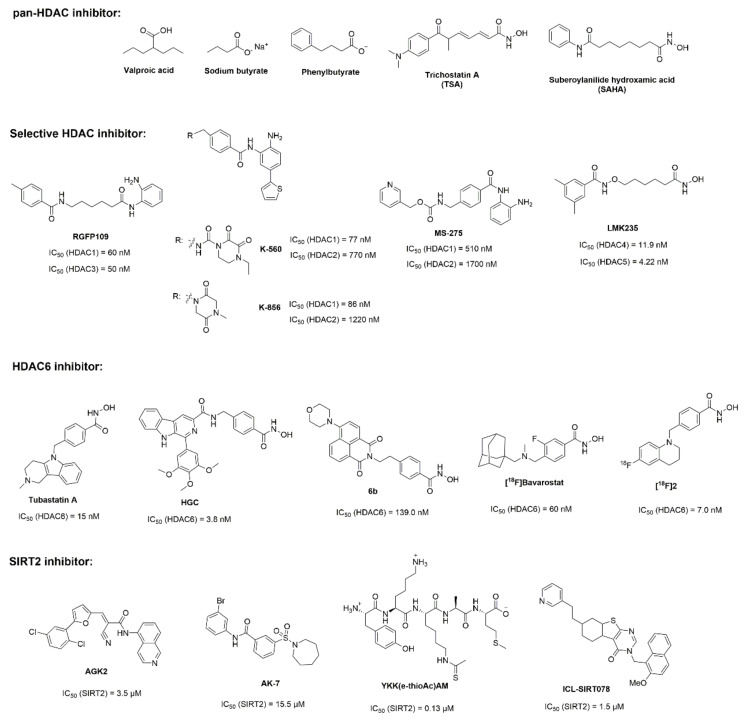
The structures of representative inhibitors of histone deacetylases used in PD study.

**Figure 3 brainsci-12-00672-f003:**
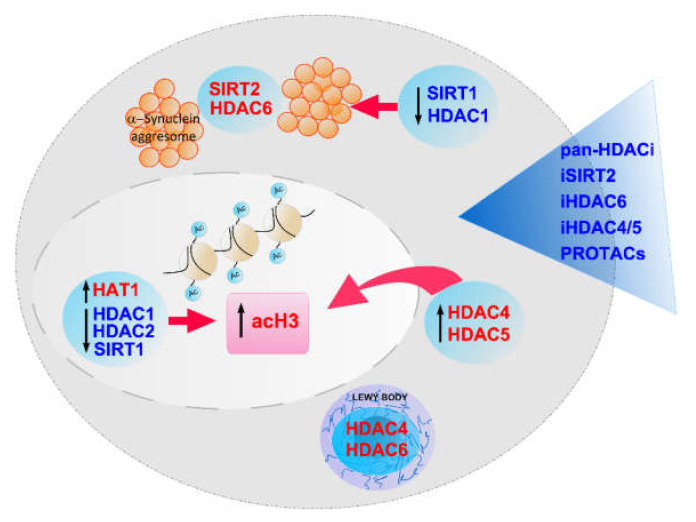
Histone deacetylases and their inhibitors discussed in this article. HDAC6, SIRT2, as well as HDAC4/5 appear to be promising targets for treating PD.

**Table 1 brainsci-12-00672-t001:** Classification and cellular localization of histone deacetylases.

Mechanism of Action	Histone Deacetylase Class	Protein(s)	Cellular Localization
Zn2+-dependent	Class I	HDAC 1, 2, 8	Nucleus
		HDAC 3	Nucleus/cytoplasm
	Class IIa	HDAC 4, 5, 7	Nucleus/cytoplasm
		HDAC 9	Nucleus/cytoplasm
	Class IIb	HDAC 6, 10	Cytoplasm
	Class IV	HDAC 11	Nucleus
NAD+-dependent	Class III	SIRT 1, 6, 7	Nucleus
		SIRT 2	Cytoplasm
		SIRT 3, 4, 5	Mitochondria

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
