# Peer review of "Histone Deacetylases as Epigenetic Targets for Treating Parkinson’s Disease"

_brainsci, 2022, doi:10.3390/brainsci12050672_

Round 1

Reviewer 1 Report

The manuscript Histone Deacetylases as Epigenetic Targets for Treating Parkinson’s Disease, is a review with a self-explanatory title: the authors outline numerous compounds that target histone deacetylases and evaluate their potential efficacy in treating Parkinson’s, or affecting related pathways.  While the manuscript is informative and contains a significant breadth and depth on the topic, the overall quality of the paper is hindered by numerous grammatical errors.  My largest recommendation for this manuscript would be significant editing.  As mentioned below, there are also some issues with redundancy in the paper.  However, the basis of the paper seems sound, and with editing, it is my opinion that the paper could be ready for publication.  Specific concerns are outlined below.

Major Concerns

While editing for grammar is needed throughout, it is especially necessary in the abstract, introduction, and further perspectives sections.

A good portion of the “further perspectives” section in repetitive with information presented earlier in the manuscript.

Moderate Concerns

Line 123: SH-SY5Y cells are mentioned without context of what type of cells they are.

Formatting issue: Figure 2 has no figure label, and is partially cut off.

A better job could be done to introduce histone acetylation and how it affects gene expression. The following statement appear in the manuscript: “Among epigenetic targets, histone deacetylases (HDACs) have been intensively investigated due to their capability of removing acetyl group from the lysine residues of histone and nonhistone substrates and thus regulating chromatin remodeling or the expression of certain genes.”  Even simply explaining first that histone acetylation is generally seen to promote gene transcription would at least provide better context to this statement.  Considering this is a paper on HDACs, this is important background.

Minor Concerns 

Your manuscript would benefit from a proper summary or conclusion at the end.

Author Response

Comment 1: “The manuscript Histone Deacetylases as Epigenetic Targets for Treating Parkinson’s Disease, is a review with a self-explanatory title: the authors outline numerous compounds that target histone deacetylases and evaluate their potential efficacy in treating Parkinson’s, or affecting related pathways.  While the manuscript is informative and contains a significant breadth and depth on the topic, the overall quality of the paper is hindered by numerous grammatical errors.  My largest recommendation for this manuscript would be significant editing.  As mentioned below, there are also some issues with redundancy in the paper. However, the basis of the paper seems sound, and with editing, it is my opinion that the paper could be ready for publication.  Specific concerns are outlined below.”

Response: We really appreciated the reviewer for recognizing that our manuscript is informative with a significant breadth and depth on the topic. And we also thank the reviewer for pointing out grammatical errors and redundancy issues. We have double-checked and corrected those errors and issues through the whole manuscript.

 Comment 2: “Major Concerns: While editing for grammar is needed throughout, it is especially necessary in the abstract, introduction, and further perspectives sections. A good portion of the “further perspectives” section in repetitive with information presented earlier in the manuscript.”

Response: We did the grammar-editing for the whole manuscript and especially in the abstract, introduction and further perspectives. Indeed, there are some repetitive information shown in the part of “further perspectives”, which is originally to summarize here what we have known about the roles and inhibitors of histone deacetylases associated with PD. Therefore, in this part, we have changed the section title from “further perspectives” to “Summary and further perspectives”. This changed title will be more matched for the contents of this section.

Comment 3: “Moderate Concerns: Line 123: SH-SY5Y cells are mentioned without context of what type of cells they are.”

Response: Thanks for pointing this out. SH-SY5Y is a human neuroblastoma cell line. We have added the corresponding context to explain this.

Comment 4: “Formatting issue: Figure 2 has no figure label, and is partially cut off.”

Response: Thanks for pointing this out. We have corrected this mistake.

Comment 5: “A better job could be done to introduce histone acetylation and how it affects gene expression. The following statement appear in the manuscript: “Among epigenetic targets, histone deacetylases (HDACs) have been intensively investigated due to their capability of removing acetyl group from the lysine residues of histone and nonhistone substrates and thus regulating chromatin remodeling or the expression of certain genes.”  Even simply explaining first that histone acetylation is generally seen to promote gene transcription would at least provide better context to this statement.  Considering this is a paper on HDACs, this is important background.”

Response: Thanks for the reviewer’s suggestion. We have added the corresponding introduction for histone acetylation and how it affects gene expression before the description of histone deacetylases showing as " In particular, histone acetylation can cause a relaxation of chromatin to allow transcriptional factor access to the DNA and thus promote gene transcription".

Comment 6: “Minor Concerns: Your manuscript would benefit from a proper summary or conclusion at the end.”

Response: Thanks for the reviewer’s suggestion, we have combined the summary and the further perspectives into the last part at the end.

Reviewer 2 Report

The review "Histone deacetylases as epigenetic targets for treating Parkinson´s Disease" describes the knowledge about HDACs in this Disorder and discusses different HDACs inhibitors as possible treatments. The review is interesting and well written. However I have some concerns:

1.- Authors focus their attention on HDAC inhibition but SIRT activation may be a target for Parkinson´s Disease. The addition of information about activators might be interesting.

2.- A table with Clinical Trials performed with Histone Deacetylase as Epigenetic targets may be informative

Author Response

Comment 1: “The review "Histone deacetylases as epigenetic targets for treating Parkinson´s Disease" describes the knowledge about HDACs in this Disorder and discusses different HDACs inhibitors as possible treatments. The review is interesting and well written. However, I have some concerns:”

Response: We are very appreciated for the reviewer favouring our manuscript. Of cause, we are happy to address the reviewer’s concerns shown as the below.

Comment 2: “1.- Authors focus their attention on HDAC inhibition but SIRT activation may be a target for Parkinson´s Disease. The addition of information about activators might be interesting.”

Response: Thanks for the reviewer raising a very good point. That’s true, SIRT activation especially for SIRT1 activation may be a target for PD, however, the development of activators is quite few and relatively difficult, compared with the development of inhibitors. In order to make the manuscript more informative, we have mentioned the activation of HDAC1 or SIRT1 could be an interesting strategy for treating PD in the corresponding sentence and the references cited in the revision. In the last part, the description is shown as “Therefore, the activation of HDAC1 and SIRT1 could be interesting strategies for treating PD [39, 72, 80, 90, 105, 112, 113].”.

Comment 3: “2.- A table with Clinical Trials performed with Histone Deacetylase as Epigenetic targets may be informative.”

Response: We agreed with the reviewer’s suggestion. If we could include a table with clinical trials associated with histone deacetylases, the manuscript will be more informative. However, since histone deacetylases as epigenetic targets are very attractive therapeutic targets for treating many human diseases, we can find hundreds of clinical studies only for active recruitment by searching “histone deacetylase” as a key term in the website of https://www.clinicaltrials.gov/ of US. Due to the page limitation, we would like to include this clinicaltrials website into the revision, which will help people who are interested in clinical studies to find out more information.

Reviewer 3 Report

The review covers an important topic, however in terms of language and grammar there are to many mistakes to list. The language needs to be corrected using the appropriate grammar and spelling correction system in MS word or similar so that all the typos mistaken times, tenses etc. which are found in almost every single line and every sentence are corrected. I also recommend proof reading by a native speaker. There are simple things such as the non-systematic use of Parkinson’s disease or the PD abbreviation throughout the manuscript which have to be accounted for but also many important statements are not properly explained, such as for example on page 9 line 285 “SIRT2 was found to be significant overexpression in PD and its regulatory substrates also involved in neurotoxicity (???) – this statement does not make sense, is not backed by the appropriate citation and spelling and grammar in the sentence is not acceptable.

Second, although the review mentions many different aspects of epigenetic research, it is necessary to bring all the various finding in a reasonable, concise and conclusive form for the reader (starting from part two (p3l84).

I suggest that the authors start with a review of the post - mortem data obtained from human Parkinson’s disease patients’ both in the sporadic and in the familiar (genetic) patients, move on in the next paragraph to the corresponding changes observed in genetic models, i.e. drosophila zebrafish or (over-expressing) transgenic mice, followed by a pargraph focusing on the toxin models ( 60HDA, MPTP and rotenone). Also the data from cell culture experiments need to be reviewed and described in a similar ystematic way.  The corresponding aspects from the various models systems have to be put together in a table for greater clarity.

Third, there has to be a separate chapter on the expression and localization of HDAC / SIRT  in the CNS and the effects on a-syn expression and aggregation and the (presumed effects in cell culture models need to be

The authors might want to identify the most important HDAC / SIRT for PD  on the basis of the current literature, identify the most conclusive data available and describe a scenario how HDAC6 and SIRT2 are implicated in PD pathogenesis.

The last chapter introducing ptroeolysis targeting chimeras / PROTAC is somewhat beyond the scope of the article and although giving an interesting outlook into some putative  future therapeutic approaches is misleading at this point.

Figure 1 is not helpful at all; there is no clear-cut indication which HDAC is localized where, in what kind of cell or cellular compartment and whether they may play a role in promoting or inhibiting accumulation and aggregation of a particular protein or are induced by certain types of (oxidative) stress or other stimuli.   

Fig. 2 is missing

Figure 3 is just a picture of various HDACS seemingly implicated or involved with a-syn aggresome but it does not become clear whether in a times clock- or counter-clockwise manner these HDAC / SIRT and / or inhibitors or activators thereof might act or interact.

Author Response

Comment 1: “The review covers an important topic, however in terms of language and grammar there are to many mistakes to list. The language needs to be corrected using the appropriate grammar and spelling correction system in MS word or similar so that all the typos mistaken times, tenses etc. which are found in almost every single line and every sentence are corrected. I also recommend proof reading by a native speaker. There are simple things such as the non-systematic use of Parkinson’s disease or the PD abbreviation throughout the manuscript which have to be accounted for but also many important statements are not properly explained, such as for example on page 9 line 285 “SIRT2 was found to be significant overexpression in PD and its regulatory substrates also involved in neurotoxicity (???) – this statement does not make sense, is not backed by the appropriate citation and spelling and grammar in the sentence is not acceptable.”

Response: Thanks for the reviewer raising very critical comments on the language and grammar as they are important for the improved quality of this manuscript. Correspondingly, we have double-checked every sentence and corrected those mistakes through the whole manuscript with the help of several English-speaking colleagues. Especially, for the term of “Parkinson’s disease” or the PD abbreviation, we have made it systematic use throughout the manuscript. And for example, the sentence on page 9 line 285 has been corrected to “SIRT2, one of Class III HDACs, was found to be significantly overexpressed in PD [40-42, 48] and its regulatory substrates are also involved in neurotoxicity and neurodegeneration [84-86].”.

Comment 2: “Second, although the review mentions many different aspects of epigenetic research, it is necessary to bring all the various finding in a reasonable, concise and conclusive form for the reader (starting from part two (p3l84).”

Response: The purpose of the part 2 in the manuscript is focusing on different isoforms of histone deacetylases involved in PD pathophysiology. The concise and conclusive form have been presented in the last part of “Summary and further perspective”.

Comment 3: “I suggest that the authors start with a review of the post - mortem data obtained from human Parkinson’s disease patients’ both in the sporadic and in the familiar (genetic) patients, move on in the next paragraph to the corresponding changes observed in genetic models, i.e. drosophila zebrafish or (over-expressing) transgenic mice, followed by a pargraph focusing on the toxin models (60HDA, MPTP and rotenone). Also the data from cell culture experiments need to be reviewed and described in a similar systematic way.  The corresponding aspects from the various model systems have to be put together in a table for greater clarity.”

Response: Thanks for the reviewer’s suggestion. In our manuscript, the logic of the organization is followed by different isoforms of histone deacetylases. As we known, there are, in general, 18 human histone deacetylases grouped into four classes. According to their catalytic mechanism, Class I, Class II and Class IV belong to zinc-dependent deacetylases (HDACs) while Class III includes nicotinamide adenine dinucleotide (NAD+)-dependent deacetylases (Sirtuins). Class I includes HDAC1, 2, 3 and 8. Class II contains two subclasses: Class IIa includes HDAC4, 5, 7 and 9 and Class IIb includes HDAC6 and 10. Class III consists of seven members, called SIRT1 to SIRT7. Class IV only has one member of HDAC11. Therefore, we prefer to organize the contents based on different histone deacetylase.

Comment 4: “Third, there has to be a separate chapter on the expression and localization of HDAC / SIRT in the CNS and the effects on a-syn expression and aggregation and the (presumed effects in cell culture models need to be”

Response: Thanks for the reviewer’s suggestion. We agreed with the reviewer’s opinion that the manuscript will be better presented if we could separate chapters. However, in order to make the logic of the manuscript consistent, we organized the contents by following the order of 18 human histone deacetylases either in part 2 or in part 3 including the expression and localization of HDAC/SIRT and those effects without separating other chapters.

Comment 5: “The authors might want to identify the most important HDAC / SIRT for PD on the basis of the current literature, identify the most conclusive data available and describe a scenario how HDAC6 and SIRT2 are implicated in PD pathogenesis.”

Response: As the reviewer mentioned, we did want to identify the most important HDAC/SIRT for PD based on the current literature. The corresponding discussion about HDAC6 and SIRT2 may be implicated in PD pathogenesis in the part of “Summary and further perspectives”. However, we cannot give such conclusions since there is still not convincing evidence to tell which isomer is critical in PD pathogenesis.

Comment 6: “The last chapter introducing ptroeolysis targeting chimeras / PROTAC is somewhat beyond the scope of the article and although giving an interesting outlook into some putative future therapeutic approaches is misleading at this point.”

Response: PROTACs as an emerging protein-degradation strategy has become prevailing and powerful tools for developing enzymatic inhibitors, which may attract more attentions for those scientists focusing on the epigenetic targets of PD and may also inspire more scientists to explore their therapeutic potentials in treating PD.

Comment 7: “Figure 1 is not helpful at all; there is no clear-cut indication which HDAC is localized where, in what kind of cell or cellular compartment and whether they may play a role in promoting or inhibiting accumulation and aggregation of a particular protein or are induced by certain types of (oxidative) stress or other stimuli.”

Response: According to the literatures, Figure 1 is to illustrate the cellular localization for different histone deacetylase. Their sub-cellular distribution may link to some PD pathogenesis although the exact roles of histone deacetylases in PD is still a puzzle. Therefore, as you can see, there are some question marks next to different localized histone deacetylases in Figure 1, indicating that we need more works to answer these questions.

Comment 8: “Fig. 2 is missing”

Response: Thanks for the reviewer pointing this out. We have corrected this mistake.

Comment 9: “Figure 3 is just a picture of various HDACS seemingly implicated or involved with a-syn aggresome but it does not become clear whether in a times clock- or counter-clockwise manner these HDAC / SIRT and / or inhibitors or activators thereof might act or interact.”

Response: Figure 3 is to summarize what we have learned the possible roles of histone deacetylases involved in PD studies and the therapeutic potentials of their inhibitors in this manuscript. To make this intention clear, we have added some explanation into the figure legend.

Reviewer 4 Report

Exciting and persuasive.

Perhaps a little about connections between the consequences of histone deacetylase blockade and PD symptoms.

Author Response

Comment 1: “Exciting and persuasive.”

Response: Thanks for the reviewer’s positive comments on our manuscript.

Comment 2: “Perhaps a little about connections between the consequences of histone deacetylase blockade and PD symptoms.”

Response: Thanks for the reviewer’s suggestion. In fact, the exact mechanism to illustrate how the inhibition of histone deacetylase connects to PD symptoms is still awaiting to explore. We did have some discussion in the final part of “Summary and further perspective”.

Round 2

Reviewer 2 Report

Authors have answered some of my concerns and my recommendation is to accept the review in the present form

Author Response

Comment: “Authors have answered some of my concerns and my recommendation is to accept the review in the present form.”

Response: We are very appreciated that the reviewer has agree to accept the current revised manuscript published on Brain Science.

Reviewer 3 Report

In the manuscript "Histone Deacetylases as Epigenetic Targets for Treating Parkinson's Disease", the authors provide a review of the literature related to HDACS in Parkinson's disease. Certainly, the revised version has been significantly improved following the reviewer's comments. Nevertheless, I believe that some important concerns still need to be addressed, such as:

  • Some members of the HDAC family are highly (and also lowly) expressed in the brain (CNS), and therefore are highly relevant to PD. Please add information about them.
  • Also, include the details about HDACs that are expressed in individual/mature neurons (e.g., HDAC4 in neuronal nuclei etc). Besides, the distinction between neuroprotective and neurotoxic HDAC categories (based on with who they interact in PD) should be mentioned.
  • Please summaries the information obtained from HDAC modification analysis (knockout, siRNA, shRNA, etc.) in context to α-Syn (and neuronal death) from PD models.
  • In context of clinical trials, authors may write a short text highlighting which HDACs have been predominantly used (alone or in combination) in PD-related trials and which have not yet been tested or should be considered. Readers should know the rational considerations behind these clinical trials aimed to induce HDAC depletion (manipulation) in the adult PD patient without any calculated side effects in the brain microenvironment.
  • Some HDACs have shown a tendency to co-localize with α-synuclein in neurons (e.g. HDAC6) and also in LB (e.g. HDAC4), please provide more information on such/similar cases.
  • Please include information about brain-penetrant HDAC inhibitor (RG2833), which showed significant information in L-DOPA-induced dyskinesia (LID).
  • Glutamate (neurotransmitter) is highly associated with Parkinson's disease, and studies have reported the neuroprotective mechanism of HDAC (TSA) preventing glutamate-induced toxicity in the medium of primary cultured astrocytes treated with MPP+ (PMID: 18628666). Please add descriptions about this.
  • I very much miss the key message of this article for readers/researchers that might help them re-think their HDAC related preclinical-clinical strategies.

Author Response

Comment 1: “In the manuscript "Histone Deacetylases as Epigenetic Targets for Treating Parkinson's Disease", the authors provide a review of the literature related to HDACS in Parkinson's disease. Certainly, the revised version has been significantly improved following the reviewer's comments. Nevertheless, I believe that some important concerns still need to be addressed, such as:”

Response: Thanks for the reviewer recognizing our revised manuscription significantly improved. Of cause, we will try further to address the reviewer’s concerns as shown as the below.

Comment 2: “Some members of the HDAC family are highly (and also lowly) expressed in the brain (CNS), and therefore are highly relevant to PD. Please add information about them.”

Response: We agree with the reviewer’s point that the expression in brain is highly relevant to PD. Therefore, we do have the corresponding description in the manuscript, e.g. “In terms of the expression levels of HDACs in PD brain samples, there are conflicting results from two separate studies. One study showed a decrease in protein levels of HDAC1, HDAC2, HDAC6, and SIRT1 in PD brain samples compared to those in controls [47]. Another study, on the contrary, demonstrated that there were no significant differences in the expression of Class I HDACs, Class II HDACs and Class III (SIRT1 and SIRT2) in PD SNpc, as compared with age-matched controls [48].”, “HDAC6 expression level was selectively increased in SN region of brain resulting in perinuclear inclusion bodies…” and “….significant overexpression of SIRT2 was found in Parkinson’s patients compared to nor-mal group [83].”

Moreover, we have added more information in this revision, showing as “Overall, HDAC1 is expressed throughout the brain and most abundantly expressed in the cerebellum while HDAC2 is expressed higher in the brain than HDAC1. However, HDAC3 is the most highly expressed in the brain among Class I HDACs and the expression of HDAC8 in brain, the only one of Class I HDACs, is still not clear. For the rest of HDACs, high expressed HDAC4-6 are found in the brain while HDAC7-10 are relatively expressed at low levels in the brain and HDAC11 remains to be disclosed [38].”

Comment 3: “Also, include the details about HDACs that are expressed in individual/mature neurons (e.g., HDAC4 in neuronal nuclei etc). Besides, the distinction between neuroprotective and neurotoxic HDAC categories (based on with who they interact in PD) should be mentioned. Please summaries the information obtained from HDAC modification analysis (knockout, siRNA, shRNA, etc.) in context to α-Syn (and neuronal death) from PD models.”

Response: We have added the information about HDAC in neurons showing as “Most HDACs are expressed in neurons. For example, HDAC1 is expressed primarily in neurons and its interacting with HDAC3 in neurons may promote toxicity while HDAC4 in neuronal nuclei appear to be associated with neuronal cell death.” The HDAC categories we have mentioned in the manuscript, e.g. “For Class III HDAC (Sirtuins), SIRT1 and SIRT5 are also known to be neuroprotective while SIRT2, SIRT3 and SIRT6 are known to be neurotoxic [40-42].” And we also add some description for HDACs since some controversial results for some isoforms showing as “According to the current acknowledge, we may safely say that HDAC1-3 and HDAC6 are neurotoxic while HDAC7, HDAC9 and HDAC10 are neuroprotective regarding to their expression in the brain.” And the intervene expression for certain isoform is a powerful tool to investigate individual function for HDACs, therefore, we have some description about these. e.g. “Silencing the expression of HDAC5 or HDAC9 by siRNAs can promote…”, “HDAC6 knockdown can effectively restore…”, etc.

Comment 4: “In context of clinical trials, authors may write a short text highlighting which HDACs have been predominantly used (alone or in combination) in PD-related trials and which have not yet been tested or should be considered. Readers should know the rational considerations behind these clinical trials aimed to induce HDAC depletion (manipulation) in the adult PD patient without any calculated side effects in the brain microenvironment.”

Response: We have added some comments in the last part showing as “Although most clinical trials for HDAC inhibitors are related to treat cancers, some clinical studies about HDAC inhibitors with carboxylic acids as ZBG groups for treating neurodegenerative diseases are ongoing [25]. As we can easily image, more selective inhibitors, multitarget HDAC-based inhibitors or PROTACs may facilitate in clinical trials for neurodegenerative diseases including PD.”

Comment 5: “Some HDACs have shown a tendency to co-localize with α-synuclein in neurons (e.g. HDAC6) and also in LB (e.g. HDAC4), please provide more information on such/similar cases.”

Response: Thanks for the reviewer’s suggestion, we do have such description as “HDAC6 was found to directly interact with α-synuclein and is involved in the Lewy bodies formation, implying that HDAC6 may relate to PD pathogenesis and progression.” And we have added the additional description as[116]. And HDAC4 may also have the similar role in LB formation [52].”

Comment 6: “Please include information about brain-penetrant HDAC inhibitor (RG2833), which showed significant information in L-DOPA-induced dyskinesia (LID).”

Response:  We have included such information as “For example, the brain-penetrant histone deacetylase inhibitor, RGFP109, attenuates L-DOPA-induced dyskinesia in the MPTP-lesioned marmoset [136].”

Comment 7: “Glutamate (neurotransmitter) is highly associated with Parkinson's disease, and studies have reported the neuroprotective mechanism of HDAC (TSA) preventing glutamate-induced toxicity in the medium of primary cultured astrocytes treated with MPP+ (PMID: 18628666). Please add descriptions about this.”

Response: We have added the descriptions about this and cited the corresponding reference as the following " Since glutamate as a neurotransmitter is highly associated with PD, there is a report for the neuroprotective mechanism of TSA preventing glutamate-induced toxicity in the medium of primary cultured astrocytes treated with MPP+ [134]".

Comment 8: “I very much miss the key message of this article for readers/researchers that might help them re-think their HDAC related preclinical-clinical strategies.”

Response: Thanks for the reviewer giving us these valuable points. We have made the corresponding summary and description in this revision.